# Is the Glass Half-Empty or Half-Full? A Mixture-Of-Tasks Perspective on Missing Modality

## Abstract

A common issue with multimodal learning setups is the unavailability of one or more modalities. Historically, missing modality has been treated as a matter of robustness, aiming to prevent performance degradation caused by stochastic loss of training and testing modalities. However, this perspective does not align with many scientific and industrial use cases of deep models where unimodal inputs are more common than having multiple modalities. Moreover, it poses practical challenges such as complicating comparisons between studies and causing ambiguity in understanding optimal model behavior. We instead propose a 'glass-half-full' approach—the Missing Modality Performance Testbed (MMPT)— which sheds light on the pivotal elements for enhancing model performance under the effect of missing modalities. MMPT reconceptualizes missing modality robustness analysis as a fundamental aspect of multimodal representation learning. This formulation allows us to connect missing modality to modality competition, an area of work that aims to improve unimodal representations in a multimodal context for late-fusion models. We create a unified framework for both missing modality and modality competition by relaxing their architectural assumptions. Via this linkage, we explore how current approaches to missing modality impact the underlying model representations and the requisite representations for favorable performance. We validate this novel perspective on a wide variety of multimodal datasets with the intention of enabling simple and clear benchmarking for future research. Finally, we present a new state-of-the-art in missing modality performance and identify potential areas for further improvement.

## 1 Introduction

Multimodal models are extending the frontiers of technological experiences on many fronts, including sophisticated education systems (Sabuncuoglu & Sezgin (2023)), intelligent robotics (Brohan et al. (2023)), effective healthcare solutions (Wang et al. (2023); Liu et al. (2023)), and interactive conversational agents (OpenAI (2023)). However, the success of these models hinges on a fallible assumption — the availability of all modalities at all times. A range of real-world constraints, such as privacy-related data sharing restrictions (Voigt & Von dem Bussche (2017)), hardware-related data collection failures, or stringent data storage policies (Voigt & Von dem Bussche (2017)), can lead to situations where modalities are unavailable. In this work, we consider Missing Completely at Random (MCAR) (Rubin (1976)), a class of missing modality problem where the probability of losing data is independent from the data itself.

In recent work, the predominant formulation of missing modality (Ma et al. (2022); Lee et al. (2023); Ma et al. (2021)) is built on robustness. Within this framework, modality loss is regarded as an adversarial condition that can drop the performance of a deep learning model. The behavior of a model is described with respect to a standard of full-modal performance where less degradation is favorable. This view is akin to describing a glass as half-empty, thereby restricting the comprehension of the missing modality challenge to describing the performance gap relative to an optimal state.

One can argue that the glass-half-empty approach naturally aligns with how humans experience their environment. Our sensory organs, such as eyes and ears, have evolved to consistently, adaptively,

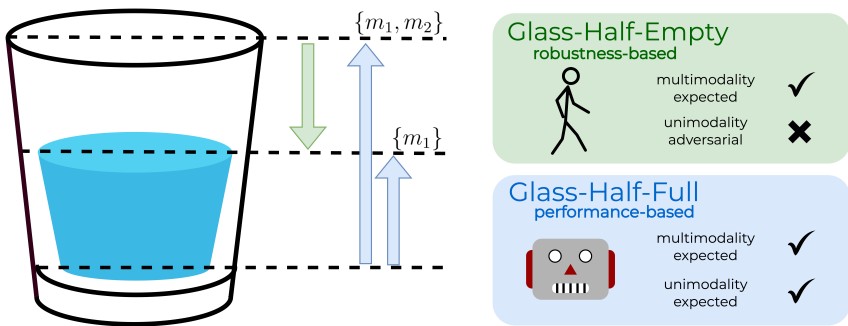

Figure 1: **Comparing the glass-half-empty and glass-half-full perspectives.** The glass-half-empty (green) perspective values the gap between multimodal $m_1, m_2$ and unimodal $m_1$ performance. The glass-half-full (blue) perspective considers the multimodal and unimodal performance separately.

and reliably capture environmental data. For example, humans can determine that water is boiling by seeing ascending pockets of air, hearing the bubbling, or touching the surface. Any loss of information from these data streams may be regarded as a challenge to decision-making, one that has the potential to lead to inaccurate conclusions. However, machines do not operate under the same principles as humans. In contrast to the human ecosystem, unimodality is not adversarial, but instead expected. Adding more modalities requires significant effort, and ensuring that we benefit off of multimodality is a monumental task all by itself.

In addition to fundamental philosophical distinctions between human and AI systems, there are practical limitations to adopting the glass-half-empty perspective. First, there is no universally-applicable way to compare the efficacy of proposed approaches. Prior studies have presented unique experiments utilizing various underlying multimodal backbones, resulting in vast performance differences across tasks. The ever-changing dynamics of experimental designs makes it difficult to fully capture the significance of novel methods. Second, the glass-half-empty perspective lacks the ability to lay down the foundational criteria that constitute optimal model performance due to an infinite number of evaluation tasks involving probabilities of modality loss. More critically, achieving higher performance in one task may compromise the performance in others, implying the absence of a universally applicable evaluation criterion by nature.

To this end, we propose treating the missing modality as a glass-half-full (see Figure 1). Instead of considering reduced performance losses of a multimodal model, we value performance increases on the task's lower and upper bounds: the aggregate unimodal and aggregate multimodal data. This perspective has been utilized briefly in other works (Ma et al. (2021); Maheshwari et al. (2023)), but we believe that its core benefit, i.e., the delineation of the fundamental tasks of missing modality, is under-explored. Additionally, this perspective sheds light on the relationship between missing modality and an entire line of work that has never been jointly considered. Specifically, modality competition (Huang et al. (2022)), also known as modality failure (Du et al. (2021)), is a phenomenon in joint training where modalities may be under-optimized. Methods proposed to address modality competition either monitor the rate of overfitting (Peng et al. (2022); Wang et al. (2019); Du et al. (2023)), or use knowledge distillation with well-trained encoders (Du et al. (2021)).

With our glass-half-full approach, we connect modality competition and missing modality under a unified framework. We demonstrate that this framework provides us with intuitions for the requisite representations to have models that behave favorably in dropped-modality scenarios. Our contributions can be summarized as follows:

- Instead of formulating missing modality as a robustness task, we connect missing modality and modality competition into the perspective of multimodal representation learning.
- Considering the multimodal model design, we separate the missing modality into two key challenges in the context of early-fusion and late-fusion models.
- We empirically discuss and present the necessary representations to construct systems that are as robust as possible to missing modality.
- We propose a novel benchmark for missing modality and evaluate a wide variety of state-of-the-art models using well-established multimodal datasets.

We provide additional details on related work in Appendix A.

Figure 2: **Visualizing the Missing Modality Performance Testbed (MMPT).** MMPT can determine missing modality performance train-time modality loss $p_{train}$ and test-time modality loss $p_{test}$. The plot on the right demonstrates missing modality performance when subject to various combinations of $p_{train}, p_{test}$ on the MM-IMDB dataset for our best late-fusion model (see Section 4.5). For extended train-time missing modality results, see Appendix B.

## 2 WHY SHOULD WE SEE MISSING MODALITY AS A GLASS-HALF-FULL?

### 2.1 THE GLASS IS HALF-EMPTY: MISSING MODALITY AS ROBUSTNESS

Consider an arbitrary classification task $\mathcal{T} = \{x_i^{m_1}, x_i^{m_2}, y_i\}, i \in \mathbb{N}$ of two modalities $\{m_1, m_2\}$, with train set $\mathcal{T}^{train}$ and identically-distributed, held-out test set $\mathcal{T}^{test}$. Let $\mathcal{T}_p^{test,m_i}$ be the missing modality task with a sample-wise probability $p$ of dropping $x_{m_i}$ in $\mathcal{T}^{test}$. For example, $\mathcal{T}_{0.1}^{test,m_1}$ represents a case where you lose $x_i^{m_1}$ with a 10-percent probability at test time and only have inputs $x_i^{m_2}, y_i$. The following is the typical glass-half-empty perspective on the robustness-based formulation of missing modality, which can be found in several previous works (Woo et al. (2023); Maheshwari et al. (2023); Lee et al. (2023); Ma et al. (2022)).

**Definition 1 (Glass-half-empty): Robustness to Missing Modality** *A model $f_\theta$'s robustness to missing modality, with respect to an appropriate evaluation metric $g(\cdot)$, is*

$$R(g, f_\theta, \mathcal{T}_p^{test,m_i}, \mathcal{T}^{test}) = g(f_\theta, \mathcal{T}_p^{test,m_i}) - g(f_\theta, \mathcal{T}^{test}). \tag{1}$$

*Here, a higher (less negative) score indicates more robustness to missing modality.*

As discussed in earlier sections, while it may seem intuitive from the glass-half-empty perspective to formulate missing modality as a robustness task, this definition carries several limitations. First, a model needs to be evaluated on an infinite number of tasks $\mathcal{T}^{test,m_i} = \{\mathcal{T}_p^{test,m_i}\}$, where $p \in [0, 1]$, to fully capture its behavior under missing modality. Since this is intractable, previous works tend to sample $p$ at fixed intervals (Ma et al. (2022); Lee et al. (2023)). Second, without prior knowledge of the target probability of modality dropout, it is unclear which tasks a model needs to train on to achieve a reasonable robustness score. A multi-task learning setup typically necessitates exposure to data from all tasks, upon which the model will be evaluated, which is impossible in our case. Previous studies avoid this problem by training on the same probability of modality dropout on which they perform evaluation (Lee et al. (2023)). Last but not least, it is possible to create an undesirable model that retains a high robustness score. As a thought experiment, let us construct a model that sabotages itself whenever both modalities are available as input. If the model performs favorably on unimodal data, then $g(f_\theta, \mathcal{T}_p^{test,m_i}) - g(f_\theta, \mathcal{T}^{test})$ becomes large and positive because the full-modal performance $g(f_\theta, \mathcal{T}^{test})$ tends to be low. Indeed, we notice that unimodal models can outperform multimodal models (see Section 4.2). This does not mean that the model is undesirable.

### 2.2 THE GLASS IS HALF-FULL: MISSING MODALITY AS A MIXTURE OF TASKS

Instead of viewing missing modality as a robustness issue, we reframe missing modality from the glass-half-full perspective by representing all possible combinations of train-time and test-time missing modality as a mixture of three tasks: the full-modal task $\mathcal{T}^{test}$ and the unimodal tasks $\mathcal{T}^{test} \setminus \{m_1\}, \mathcal{T}^{test} \setminus \{m_2\}$ where $\mathcal{T}^{test} \setminus \{m_i\} = \mathcal{T}_1^{test,m_i}$.

**Definition 2 (Glass-half-full): Missing Modality Performance Testbed (MMPT)** *A model $f_\theta$'s missing modality performance profile, with respect to an appropriate evaluation metric $g(\cdot)$, is*

$$\text{MMPT}(g, f_\theta, \mathcal{T}^{test}, \mathcal{T}^{test}\backslash\{m_i\}) = (g(f_\theta, \mathcal{T}^{test}), g(f_\theta, \mathcal{T}^{test}\backslash\{m_2\}), g(f_\theta, \mathcal{T}^{test}\backslash\{m_1\}))\,. \quad (2)$$

We argue that this formulation comprehensively captures any combination of training and testing probabilities of modality loss. For test-time modality loss, we evaluate on $\mathcal{T}_p^{test,m_i}$ by performing stochastic logit fusion: for each sample, we take the logits of $f_\theta$ on $\mathcal{T}^{test} \setminus \{m_i\}$ with probability $(1-p)$ and $\mathcal{T}^{test}$ with probability $p$. For train-time modality loss, we collect all the available unimodal data by splitting the paired data and redistributing it with the unpaired inputs. Then, we train a model on the aggregate unimodal and multimodal data. Figure 2 shows how MMPT naturally captures any combination of train and test-time probabilities of modality dropout. The plot at the right shows performance on how a well-trained model behaves when subject to $p_{train}$ percent probability of train-time text dropout, and $p_{test}$ percent probability of test-time text dropout. As the amount of aggregate unimodal image data does not change depending on $p_{train}$, the model has approximately the same performance on $\mathcal{T}^{test} \setminus \{text\}$ (where $p_{test} = 100\%$ in our plot). In contrast, we have less multimodal data as $p_{train}$ increases so the model loses performance on $\mathcal{T}^{test}$ (where $p_{test} = 0$ in our plot).

## 3 EMBRACING THE GLASS-HALF-FULL VIEW ON MISSING MODALITY

We develop a unified perspective of missing modality and modality competition. With this analogy, we aim to gain insight into the representations that result in favorable performance when modalities are stochastically absent during training or testing phases.

### 3.1 CONNECTING MISSING MODALITY WITH MODALITY COMPETITION

We identify modality competition (Huang et al. (2022)) in our formulation of missing modality. Modality competition posits that weaker modalities may be under-optimized during joint training of late-fusion models, which may impact multimodal performance. Let us consider again a multimodal classification task $\mathcal{T} = \{x_i^{m_1}, x_i^{m_2}, y_i\}_i$. A standard late-fusion model processes $\mathcal{T}$ as follows. First, the input is passed through the respective encoder to obtain modal representations $z_i^{m_1} = \phi^{m_1}(x_i^{m_1})$ and $z_i^{m_2} = \phi^{m_2}(x_i^{m_2})$. Then, $z_i^{m_1}, z_i^{m_2}$ are concatenated and passed to a linear classification head:

$$\boldsymbol{y}_i^{pred} = \boldsymbol{W}\text{concat}([z_i^{m_1}, z_i^{m_2}])^T + \boldsymbol{b}$$

After training, each encoder is trained with a newly-initialized head and linear-probed independently. However, when one of the encoders achieves nearly perfect performance on the training dataset, the other modality may significantly lag behind (Wang et al. (2019); Huang et al. (2022)). At this stage, the model will not optimize its encoders any longer as its training loss has converged. We denote the lagging modality as the "weaker" modality, and the overfitting modality as the "stronger" modality. In order to empirically measure the strength of a modality, we define the following weakness score.

**Definition 3: Weakness Score** *The weakness score of modality $m_1$ is*

$$W(\phi^{m_1}, \phi^{m_2}, \mathcal{T}^{train}\backslash\{m_1\}, \mathcal{T}^{train}\backslash\{m_2\}) = l(\phi^{m_1}, \mathcal{T}^{train}\backslash\{m_2\})/l(\phi^{m_2}, \mathcal{T}^{train}\backslash\{m_1\}) \quad (3)$$

*where $\phi^{m_1}, \phi^{m_2}$ are encoders for modality $m_1, m_2$ respectively and $l$ is the cross-entropy loss.*

A weakness score less than 1 indicates that $m_1$ is dominant, whereas a weakness score greater than 1 indicates that $m_2$ is dominant.

### 3.2 UNIFYING MISSING MODALITY AND MODALITY COMPETITION

In both the glass-half-full view of missing modality and modality competition, the aim is to enhance performance on unimodal and multimodal tasks. Additionally, modality competition identifies a unique problem that hinders this objective. In order to achieve a profound understanding of how modality competition fits into missing modality and vice versa, it is important for us to consolidate the architectural assumptions for each. Many prior works have examined missing modality using various fusion schemas (Ma et al. (2022; 2021)), whereas modality competition has solely been explored in the context of late-fusion models (Huang et al. (2022); Wang et al. (2019)).

First, let us consider late-fusion models for which both missing modality and modality competition have been studied. The only difference between how models are evaluated on unimodal and multimodal tasks is an additional linear probing step that modality competition requires. In practice, we note that linear probing can sometimes provide slightly better performance, especially for models trained on a multi-task objective, so we adopt this step for missing modality as well. In summary, there are three steps for evaluating a late-fusion model on MMPT. First, we train on the aggregated unimodal and multimodal data. Second, we linear-probe the unimodal encoders $\phi^{m_1}, \phi^{m_2}$ to evaluate performance on $\mathcal{T}^{test} \setminus \{m_i\}$. Third, we linear-probe the concatenated encoder outputs to evaluate performance on $\mathcal{T}^{test}$.

Next, we consider early-fusion models. Extending the concept of modality competition to this fusion schema is not straightforward because of the blending of representations across different modalities. Although there isn't a general solution to this problem for all types of early-fusion models, we propose a method to unify modality competition and missing modality for early-fusion transformers.

Early-fusion multimodal transformers learn from multimodal representations through self-attention. The self-attention mechanism can be decomposed into two types: modality self-attention where inputs from the same modality attend to each other, and cross-attention where inputs from one modality attend to those from the other. However, cross-attention does not apply to early-fusion model at test-time on unimodal data. Therefore, we propose a method inspired by Ma et al. (2022) to decouple the unimodal components, where we modify the self-attention mechanism as

$$\text{softmax}\left(\frac{QK^T}{\sqrt{d_e}} + M_i\right) V \rightarrow \text{softmax}\left(\frac{QK^T}{\sqrt{d_e}} + M_i + M_c\right) V,$$

where the mask $M_i \in \{0, -\infty\}^{d_e \times d_e}$ removes padding from the attention mechanism, and the mask $M_c = \begin{bmatrix} 0 & -\infty \\ -\infty & 0 \end{bmatrix} \in \{0, -\infty\}^{d_e \times d_e}$ removes cross-attention between modalities.

After adding the $M_c$ mask, the final hidden state can be interpreted as $h^{final} = \text{concat}([h^{m_1}, h^{m_2}])$, where $z^{m_1}, z^{m_2} \in \mathbb{R}^{d_h \times l^{m_i}}$, $d_h$ being the embedding dimension and $l_{m_i}$ the number of tokens associated with modality $m_i$. Then, we can extract representations by average-pooling across the length dimension, i.e., $z^{m_i} = \text{avg pool}(h^{m_i})$. Since we separated unimodal and multimodal components, we are now able to evaluate early-fusion models similarly to late-fusion models on MMPT.

---

*Architecture-independent Pipeline for MMPT Evaluation*

1. Train a system on the aggregated unimodal and multimodal data.

2. Split the model into unimodal components $\psi^{m_1}, \psi^{m_2}$ and multimodal component $\psi^{m_1, m_2}$.

3. Linear-probe the unimodal components $\psi^{m_1}, \psi^{m_2}$ to evaluate on $\mathcal{T}^{test} \setminus \{m_i\}$.

4. Linear-probe the multimodal component $\psi^{m_1, m_2}$ to evaluate on $\mathcal{T}^{test}$.

---

By decomposing an early-fusion model into unimodal and multimodal parts, we can directly apply training methods from modality competition and evaluate modality weakness by setting $\phi^{m_i} = \psi^{m_i}$.

# 4 AN ARSENAL OF METHODS FOR IMPROVING UNIMODAL AND MULTIMODAL FEATURES

In the following, we evaluate our methods for multimodal feature learning on MMPT and present our main findings. We use $100\%$ of the training data to identify the upper bound in missing-modal performance capabilities. We report results with state-of-the-art pre-trained backbones, which are not considered by most recent works on modality competition (Huang et al. (2022); Peng et al. (2022); Du et al. (2021; 2023)). This is significant because the relatively large parameter count of state-of-the-art models causes different overfitting behaviors on the data.

We consider a diverse range of 2-modality classification datasets for our experiments: MM-IMDB (Arevalo et al. (2017)), N24News (Wang et al. (2022b)), UPMC Food101 (Ignazio Gallo & Grassa (2020)), Crisis-MMD (Ofli et al. (2020); Alam et al. (2018)) for vision-language, and AV-MNIST (Vielzeuf et al. (2018)) for audio-visual. We summarize each of these datasets in Appendix C.1.

Table 1: **Arsenal of Methods on MMPT on Common Multimodal Datasets for Late Fusion**
Results on the six datasets on Both ($\mathcal{D}$) , Text/Audio ($\mathcal{D} \setminus \{x^{image}\}$), and Image
($\mathcal{D} \setminus \{x^{text/audio}\}$). Best performance is bolded, methods that outperform unimodal experts are
colored in blue. All datasets are evaluated using Macro F1.

| | MM-IMDB | | | N24News Headline | | | N24News Body | | |
|---|---|---|---|---|---|---|---|---|---|
| | Both | Text | Image | Both | Text | Image | Both | Text | Image |
| **Naive** | 59.3 | 59.9 | 24.1 | 78.2 | 71.5 | 49.9 | 87.3 | 85.8 | 49.4 |
| **Unimodal** | - | 60.8 | **35.5** | - | 71.8 | 51.0 | - | 86.1 | 51.1 |
| **Dropout** | **59.7** | 60.2 | 27.5 | 76.7 | 70.7 | **51.1** | 85.8 | 85.8 | 50.0 |
| **MTL** | 56.9 | 59.1 | 32.5 | **78.8** | 72.0 | **51.1** | 87.8 | **86.6** | 50.4 |
| **G-Blend** | 58.9 | 60.2 | 33.0 | 77.3 | 71.9 | 50.7 | 85.3 | 85.8 | 50.9 |
| **OGM-GE** | 59.5 | 58.9 | 32.1 | 78.1 | 71.8 | 50.3 | 86.9 | 85.3 | 51.2 |
| **UMT** | 59.6 | 61.1 | 35.2 | 75.2 | 72.1 | 50.8 | 83.5 | 85.3 | **51.4** |

| | Crisis-MMD | | | UPMC Food101 | | | AV-MNIST | | |
|---|---|---|---|---|---|---|---|---|---|
| | Both | Text | Image | Both | Text | Image | Both | Audio | Image |
| **Naive** | 42.0 | 36.1 | **34.2** | 93.2 | 85.0 | 67.1 | 90.7 | 86.1 | 48.9 |
| **Unimodal** | - | 37.8 | 33.9 | - | **86.4** | 72.0 | - | 85.0 | **49.5** |
| **Dropout** | 41.6 | 37.2 | 31.5 | 91.2 | **86.4** | 69.4 | **91.7** | 85.3 | 48.2 |
| **MTL** | 39.5 | 37.9 | 33.7 | **93.8** | 86.3 | **72.1** | **91.7** | 85.3 | 48.4 |
| **G-Blend** | 41.0 | 36.6 | 33.3 | 93.5 | 86.1 | 68.5 | 91.3 | 85.5 | 48.2 |
| **OGM-GE** | 40.4 | 37.0 | 32.9 | 93.6 | 86.0 | 71.1 | 91.6 | **86.2** | 48.3 |
| **UMT** | **42.9** | 37.6 | 33.4 | 93.6 | **86.4** | 71.7 | 91.6 | 86.0 | 48.3 |

## 4.1 TRAINING STRATEGIES

We identify recent training strategies and baselines developed for missing modality and modality
competition. In the following, we use superscript [1] for methods originating from missing modality
literature and superscript [2] for methods originating from modality competition literature.

**Naive Joint Training Baseline (Naive):** Naive joint-training baseline is a model trained on $\mathcal{T}^{train}$.
For late-fusion, we fuse the Bidirectional Encoder Representations from Transformers (BERT) (De-
vlin et al. (2019)) for text modalities, the Vision Transformer (ViT) (Dosovitskiy et al. (2021)) for
image modalities, and the Audio Spectrogram Transformer (AST) (Gong et al. (2021)) for audio
modalities. For early-fusion modeling, we use the Vision-and-Language Transformer (ViLT) (Kim
et al. (2021)) for the vision-language tasks and the Textless Vision-Language Transformer (TVLT)
(Tang et al. (2022)) for the audio-visual tasks.
**Unimodal Baseline (Unimodal):** The unimodal baseline is a model trained on $\mathcal{T}^{train} \setminus \{m_i\}$. We
use the same pretrained models as in the naive joint training baseline. For the early-fusion case, we
modify the attention masks as discussed to obtain unimodal components.
**Modality Dropout[1] (Dropout) (Xiao et al. (2020); Ma et al. (2022)):** With an empirically-chosen
probability of $p = 0.5$, we drop modality $m_i$ at training time. We find that dropping both modalities
at once can cause training instability, so instead we train a separate model for each modality. The
reported multimodal performance is the maximum of the two.
**Multi-task Learning[1] (MTL):** Inspired by Ma et al. (2022), we train a model on the multimodal
task $\mathcal{T}$ and the two unimodal tasks $\mathcal{T}^{train} \setminus \{m_i\}$ by taking a linear combination of the losses. In
late fusion, the objective function is $L = L_{multimodal} + \alpha L_{text} + \beta L_{image}$. In early fusion, we
obtain the unimodal representations as described in Section 3.2. The objective function is the same.
**Gradient Blending[2] (G-Blend) (Wang et al. (2019)):** G-blend works in two steps. In the first step,
the overfitting rate for each modality is estimated by comparing the encoder checkpoints after a fixed
number of epochs. Then, the multimodal model is trained for the same number of epochs according
to the generalization-to-overfitting ratio.
**Online Gradient Modulation[2] (OGM-GE) (Peng et al. (2022)):** OGM-GE computes a discrep-
ancy ratio of the contribution of each modality to the multimodal prediction as a function of the
output logits of the model. Then, the learning rate of each encoder is changed based on the discrep-
ancy ratio so modalities do not overfit too quickly.

Table 2: **Arsenal of Methods on MMBT on Common Multimodal Datasets for Early Fusion**
Results on the six datasets on Both ($\mathcal{D}$) , Text/Audio ($\mathcal{D} \setminus \{x^{image}\}$), and Image
($\mathcal{D} \setminus \{x^{text/audio}\}$). Best performance is bolded, methods that outperform unimodal experts are colored in blue. All datasets are evaluated using Macro F1.

| | MM-IMDB | | | N24News Headline | | | N24News Body | | |
| --- | --- | --- | --- | --- | --- | --- | --- | --- | --- |
| | Both | Text | Image | Both | Text | Image | Both | Text | Image |
| **Naive** | 56.0 | 41.2 | 26.3 | **72.9** | 32.3 | 49.1 | **82.7** | 68.5 | 46.2 |
| **Unimodal** | - | 50.2 | 34.7 | - | 59.2 | **52.3** | - | 79.8 | 51.2 |
| **Dropout** | 54.5 | 44.6 | 31.2 | 71.1 | 51.3 | 48.8 | 81.1 | 69.9 | 48.1 |
| **MTL** | **56.4** | 50.4 | 34.0 | 70.8 | 58.6 | 51.1 | 82.0 | 78.1 | **51.9** |
| **G-Blend** | 54.9 | 47.7 | 34.6 | 71.9 | 58.6 | 49.7 | 81.7 | 77.1 | 49.3 |
| **OGM-GE** | 55.7 | 45.9 | 31.9 | 72.2 | 52.2 | 42.1 | 80.9 | 76.3 | 49.9 |
| **UMT** | 54.4 | 50.9 | 34.9 | 71.2 | 59.9 | 50.3 | 82.5 | 79.9 | 50.2 |

| | Crisis-MMD | | | UPMC Food101 | | | AV-MNIST | | |
| --- | --- | --- | --- | --- | --- | --- | --- | --- | --- |
| | Both | Text | Image | Both | Text | Image | Both | Audio | Image |
| **Naive** | **43.9** | 26.4 | 33.5 | 92.4 | 64.1 | 62.0 | 81.3 | 65.1 | 47.9 |
| **Unimodal** | - | **36.2** | 35.0 | - | 86.4 | 69.8 | - | 73.5 | 50.3 |
| **Dropout** | 39.6 | 27.5 | 37.9 | **93.1** | 74.8 | 64.3 | 80.4 | 72.6 | 49.9 |
| **MTL** | 40.5 | 30.6 | **38.6** | 93.0 | 85.9 | 68.8 | 81.1 | 74.2 | 50.2 |
| **G-Blend** | 40.4 | 31.4 | 38.2 | 92.9 | **86.9** | 67.3 | 81.3 | 72.8 | 49.8 |
| **OGM-GE** | 39.9 | 28.6 | 37.3 | 92.8 | 79.1 | 66.9 | 81.1 | 72.1 | 50.0 |
| **UMT** | 42.0 | 35.9 | 38.1 | 92.8 | 86.6 | **70.1** | 81.2 | 74.0 | **50.3** |

**Uni-Modal Teacher**[2] **(UMT) (Du et al. (2021)):** The naive joint-training objective is augmented with a knowledge-distillation loss from each of the encoders to the frozen unimodal baseline models. In particular, the additional loss functions use the square of the L2 distance between the hidden states of the encoders and the frozen unimodal models $l_{distill} = \|h_{enc} - h_{baseline}\|_2^2$. For early-fusion, the unimodal encoders are the masked version of ViLT and TVLT.

## 4.2 Evaluating Late Fusion Models on MMPT

We compare different late fusion models on MMTP in Table 1.

*On Unimodal Performance*: We observe that all five training strategies in Table 1 outperform naive-joint training on unimodal tasks, apart from a few exceptions (e.g., the image modality for Crisis-MMD for late-fusion). However, there are certain appeals to multi-task learning in other dimensions than performance. Compared to modality dropout, there is more training stability. Compared to UMT, there are 2x fewer trainable parameters. Compared to OGM and G-blend, the training speed is considerably faster. We also notice that none of the training strategies consistently surpass the unimodal experts, while in many cases they converge to about the same performance as the unimodal counterparts. This phenomenon is anticipated given the nature of the late-fusion design — the unimodal baseline is a well-trained version of the encoder that can address the modality competition gap, allowing the model to reach similar performance compared to the unimodal baseline.

*On Multimodal Performance*: The multimodal results (Table 1) indicate that both modality competition and missing modality training strategies improve multimodal performance. However, we note that this performance improvement is often marginal in comparison to that of naive joint training.

*An Overflowing Glass — The Failure of Robustness*: It is common knowledge within the modality competition domain that unimodal experts can outperform multimodal models. Our results reaffirm this: for the MM-IMDB dataset, we see that the best text expert (Macro F1: 61.1%) outperforms the best multimodal expert (Macro F1: 59.7%). As unimodality is more favorable than multimodality in this case, treating missing modality as a robustness task does not seem suitable.

## 4.3 Evaluating Early Fusion Models on MMPT

We compare different early fusion models on MMTP in Table 2.

*On Unimodal Performance*: For early-fusion, it is unclear to what extent modality competition explains the gap between naive joint-training and well-performing unimodal experts. Regardless, we still notice consistent improvement over unimodal task performance for all the training strategies. Furthermore, similar to late-fusion, our highest-performing models seem to perform only marginally better than separately-trained unimodal experts. We believe the efficacy of the approaches can be attributed to having some notion of unimodal performance in the objective function.

*On Multimodal Performance*: In contrast to late fusion results, the glass-half-full approach does not consistently yield higher multimodal performances than naive joint training on MMPT, even for models with significantly higher unimodal performance. This observation contradicts with the modality competition theory, which states that addressing the under-optimization of unimodal features leads to improved multimodal performance. Instead, we observe that naive joint-training yields higher mulitmodal performance in many datasets.

*On the Gap between Late and Early Fusion Performance:* Across the datasets, we observe a gap between early-fusion and late-fusion performance. While the unimodal image performance of ViLT rivals that of late-fusion, the unimodal text performance trails behind. Similarly, the unimodal image performance of TVLT is similar to ViT, but for the audio performance, TVLT lags behind AST. One plausible explanation for this is pretraining weight initialization: ViLT is initialized with ViT, and TVLT is initialized with ViT-MAE (He et al. (2022)). Specifically, we believe a proper weight initialization of early-fusion models can make their performance rival late-fusion across all datasets.

### 4.4 MMPT UNCOVERS MODALITY WEAKNESS

Table 3: **Modality Weakness and Unimodal Performance Improvements**
Text/audio weakness score of naive joint training and improvements of best-performing approach over naive joint training. A weakness score below one means text/audio is dominant, and a score above one means image is dominant.

| | MM-IMDB | N24News Headline | N24News Body | Crisis-MMD | UPMC Food101 | AV-MNIST | |
|---|---|---|---|---|---|---|---|
| **Text/Audio Weakness Score** | 0.483 | 0.392 | 0.128 | 1.916 | 0.713 | 0.706 | late |
| **Text/Audio Improvement** | 1.2 | 0.6 | 0.8 | 1.3 | 1.4 | 0.1 | |
| **Image Improvement** | 11.4 | 1.2 | 2.0 | -0.3 | 5.0 | 0.6 | |
| **Text/Audio Weakness Score** | 0.638 | 1.562 | 0.524 | 1.302 | 1.422 | 0.322 | early |
| **Text/Audio Improvement** | 9.7 | 27.6 | 11.4 | 9.8 | 22.8 | 9.1 | |
| **Image Improvement** | 8.6 | 3.2 | 5.7 | 5.1 | 8.1 | 2.4 | |

For late-fusion models, modality weakness correlates with the potential for unimodal improvement. We observe for the one dataset with a greater-than-one weakness score (Crisis-MMD) we have more text improvement than image. For all other datasets, there is significantly more image improvement than text. Furthermore, we note that N24news-body has a smaller weakness score than N24headline, which suggests that the image modality is weaker in N24news-body. This agrees with our intuition because although the image data between both datasets is the same, there is significantly more text in N24news-body. On top of this, we notice that the amount of unimodal image improvement in N24news-body is more than in N24news-text (2.0 vs 1.2). However, in general, we do not notice a strong relationship between unimodal improvement and the magnitude of the weakness score.

For early-fusion models, we do not observe a correlation between weakness score and image versus text improvement. This provides evidence that modality competition either does not exist for early-fusion models or is confounded with other effects that cause poor unimodal performance.

### 4.5 MMPT IDENTIFIES THE STATE-OF-THE-ART PERFORMANCE ON MISSING-MODALITY

One benefit of MMPT's separation of unimodal and multimodal tasks is that individually selecting the best models for each task leads to the best-performing system overall. Figure 3 shows the performance of our best model compared to two recent papers on missing modality (Ma et al. (2022); Lee et al. (2023)) on the MM-IMDB dataset and the naive-training baseline. Our figure shows that we consistently outperform previous work across all percentages of test-time modality loss.

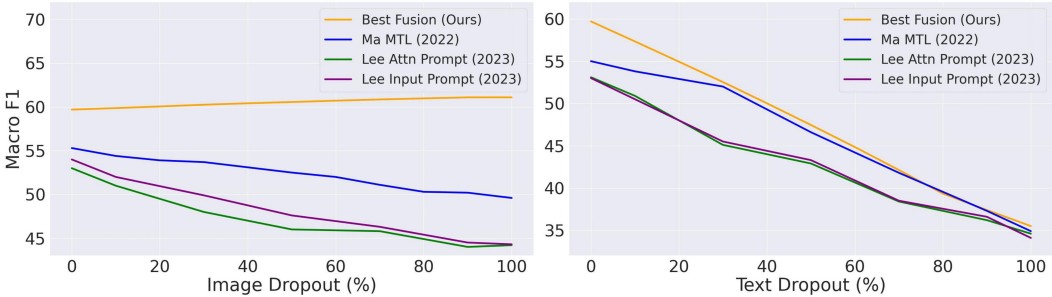

Figure 3: **Best-fusion in relation to recent work.** Comparison between Best-Fusion (Section 4.5) and recent missing-modality work (Ma et al. (2022); Lee et al. (2023)) which evaluate on MM-IMDB. Performance on 100% train modality availability and $p$ test modality dropout in intervals of 10%. Image is dropped in the left plot, and text is dropped in the right plot.

## 5 BEYOND GLASS-HALF-FULL: CAN WE IMPROVE UNIMODAL EXPERTS?

Table 4: **Feature-level imputation** with our naive and best models with respect to the unimodal baseline (see Section 4)

|  | Late-Fusion | | | Early-Fusion | | |
|---|---|---|---|---|---|---|
|  | **Unimodal** | **LF Naive** | **LF Best** | **Unimodal** | **EF Naive** | **EF Best** |
| **Text/Audio** | 61.1 | 59.9 | 60.7 | 50.2 | 40.1 | 52.9 |
| **Image** | 35.5 | 36.3 | 38.4 | 34.7 | 21.2 | 36.8 |

Through our experiments, we established that unimodal experts trained solely on unimodal data are difficult to beat consistently. However, previous missing modality works have found methods to improve on their performance by significant margins: data imputation (Ma et al. (2021)). Here, we highlight the importance of addressing modality competition before feature-level data imputation.

We train a variational autoencoder with standard ELBO loss function:

$$L = -\mathbb{E}_{q(z|x^{m_1})}[\log p(x^{m_2}|z)] + D_{KL}(q(z|x^{m_1})||p(z))$$

where $x^{m_1}$ is the output of the unimodal component for the modality you have available and $x^{m_2}$ is the out for the unimodal component for the modality being imputed. Table 4 shows the imputation results using 4 different unimodal components (naive early-fusion, best early-fusion, naive late-fusion, and best late-fusion) on the unimodal task $\mathcal{T} \setminus \{m_i\}$. We also report a baseline of unimodal experts with no imputation. Our results highlight the importance of having strong unimodal experts even before imputation, as imputing with our best models performs better most of the time compared to imputing with our naive models. In fact, we note that for early-fusion, imputing with the naive models lowers performance, leading to worse performance than unimodal experts.

## 6 CONCLUSION

We challenge the robustness-based formulation of missing modality and offer a novel glass-half-full perspective. Motivated by this view, we develop the Missing Modality Performance Testbed (MMPT), which identifies the core tasks to create models robust to missing modality. We use MMPT to unify missing modality with modality competition under a common light of multimodal representation learning by navigating through their assumptions on fusion strategies. Then, we benchmark state-of-the-art models on MMPT using diverse multimodal datasets and training methods from both bodies of work with the aim of enabling comparisons between current and future work. For both late and early fusion models, we discover that missing modality and modality competition approaches offer significant improvements over naive joint-training on unimodal tasks. However, we do not consistently see multimodal performance improvements. Finally, we demonstrate data imputation with well-trained encoders can push the boundaries on unimodal expert performance.

## 7    REPRODUCIBILITY STATEMENT

To promote reproducibility, we provide the code to our models and training scripts in the supplementary materials. We note that all the datasets we have used in our experiments are publicly available except AV-MNIST, which can differ from other publicly available instances of the dataset because of noise-injection (see Appendix C.1). Thus, we attach the AV-MNIST dataset in our supplementary materials as well. All pre-trained weights are from Hugging Face's model zoo, and we point to the specific checkpoints we have used in Appendix C.2. All evaluation results with stochastic missing modality probabilities have been averaged over 100 seeds.

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

## A   RELATED WORK

**Multimodal Learning** Multimodal learning aims to exploit complementary information between modalities to improve performance in downstream tasks. Recently, transformers have been shown to achieve state-of-the-art performance across many multimodal tasks . There are several paradigms to fuse the information from the inputs. In early fusion (Kim et al. (2021); Tang et al. (2022)), modalities are concatenated before they are passed into the model. In late fusion, individual encoders first process each modality and combine their representations afterward (Wang et al. (2019); Huang et al. (2022)). There are also mid-fusion models (Wang et al. (2022a); Xu et al. (2023); Li et al. (2023)), which combines both early and late-fusion with large modality-specific encoders and large modality-interaction modules.

**Missing Modality** There has been an increasing interest in designing models which are robust to missing modalities. Most papers consider pairs of modalities, where there is a probability of one of the two to be lost (Ma et al. (2022; 2021); Lee et al. (2023); Woo et al. (2023); Maheshwari et al. (2023); Zeng et al. (2022); Zhao et al. (2021)). Several papers have studied this problem in the late-fusion domain (Ma et al. (2021)) whereas more recently, works have considered early-fusion models (Kim et al. (2021)) as well. The predominant strategies to combat missing modality are exposing the model to missing-modal data during train time (Ma et al. (2022); Lee et al. (2023)) and imputing the missing data (Ma et al. (2021); Woo et al. (2023)).

**Modality Competition** Modality competition, also known as modality failure, addresses suboptimal representations caused by different rates of modality overfitting in multimodal models. Modality

Figure 4: **Extended Train/Test-Time Missing Modality Results:** Using MMPT, we show different permutations of train-time, test-time modalities across various probabilities for modality dropout on the MM-IMDB dataset.

competition was initially proven with shallow networks (Wang et al. (2019); Huang et al. (2022)). More recently, there have been attempts to address modality competition in deeper models. There are a variety of approaches proposed to address modality competition. Wang et al. (2019) directly estimate the rate of overfitting by comparing future model checkpoints to previous ones. A couple methods (Shen et al. (2023); Peng et al. (2022)) strive to balance the overfitting rates by modifying the gradients. Other methods modify the objective function directly by adding proxies for modality overfitting (Du et al. (2021; 2023)).

## B   MORE ON TRAINING-TIME MISSING MODALITY

With MMPT, we can illustrate the behavior of well-trained models for any mixture of training-time missing modality and evaluation-time modality as shown in Figure 4. We define the mixture of missing modalities as the logit fusion between the model's logits on full-modality and that of the missing modality. Naturally, this gives us 4 different combinations of logit fusions when we have two modalities in a given dataset.

Without the loss of generality, if we are missing the modality $m_1$ with a probability $p_{train}$ during training-time and missing the modality $m_2$ with a probability $p_{test}$ during test-time, we can express this behavior by taking the logits of the model trained on $(1 - p_{train})$ of the modalities $m_1$ and $m_2$ at $(1 - p_{test})$ and the logits of the model trained on $(1 - p_{train})$ of the modality $m_1$ at $p_{test}$. For example, if $p_{train} = 90\%$ for the text modality and $p_{test} = 60\%$ for the image modality, we can expect to test the model's multimodal capabilities on text and image for the 40% of the test-time trained on 10% of multimodal data and unimodal capabilites 60% of the time trained on 10% of the entire text when image is missing. Therefore, we take the logits of the model trained on both text and image at 10% at 40% probability and take the logits of the model trained at 10% of only text at 60% probability. The result of experiments for the train-time and test-time modality differing with

varying probabilities is shown in the top right and bottom left plot of Figure 4. The top left and bottom right plot of Figure 4 shows a case when the missing modality for train-time and test-time is the same, where we take the unimodal logits of the non-missing modality trained at 100% training data and fuse it with multimodal data trained with $(1 - p_{train})\%$ of total multimodal training data.

## C  Experiment Details

### C.1  Dataset Details

**MM-IMDB (Arevalo et al. (2017)):** The MM-IMDB dataset is a collection of images of movie posters and corresponding textual plot descriptions with the goal of predicting the movie genre. It is a 23-class multilabel classification task with 25,956 paired samples.

**N24News (Wang et al. (2022b)):** N24News has 61,218 samples of paired text and image from New York Times articles. The task is to predict one of the 24 categories (genres) of news. There are different sources of text available: headline, caption, abstract, and body. For this work, we run experiments on both the headline and the body as they contain different volumes of text.

**UPMC Food101 (Ignazio Gallo & Grassa (2020)):** The UPMC Food101 dataset contains 100,000 pairs of food recipe titles and images, with the goal of predicting one of 101 different types of food. We note that the text modality is strong for the dataset, since many text samples contain the name of the food itself.

**Crisis-MMD (Ofli et al. (2020); Alam et al. (2018)):** CrisisMMD is a twitter dataset that contains 18085 samples of paired tweets and images collected during seven national disasters. CrisisMMD has 3 different tasks, and we evaluate on the humanitarian task where we want to predict one of eight different humanitarian categories.

**AV-MNIST (Vielzeuf et al. (2018)):** AV-MNIST, or Audiovisual MNIST, consists of 70,000 pairs of written and spoken digits. The images are based on the MNIST dataset (LeCun et al. (2010)), and we disturb the images by PCA projection with only 25% of the energy, identical to what the original authors (Vielzeuf et al. (2018)) suggest. In the audio samples, the spoken digits from the Free Spoken Digit Database (Jackson et al. (2018)) are mixed with the randomly chosen noise sounds from the ESC-50 dataset (Piczak (2015)) with a noise power of 0.5. These distortions are included, as they yield around 99% accuracy in almost all cases without such manipulation. The audio samples are resampled to 16 kHz to utilize a pre-trained AST model. As no official code to generate the AV-MNIST dataset is publicly available, we refer to the following code to create the dataset, except that we output raw waveforms rather than spectrograms: `https://github.com/slyviacassell/_MFAS/blob/master/datasets/avmnist_gen.py` We then split the dataset, 60,000 samples for the train set, and 5,000 for the validation and test set, respectively. We include AV-MNIST to diversify our results past the vision-language domain.

### C.2  Training Details

We utilized the following pre-trained models from Hugging Face[1] for each modality: `bert-base-uncased`[2] for BERT, `google/vit-base-patch16-224`[3] for ViT, and `MIT/ast-finetuned-audioset-10-10-0.4593`[4] for AST. For early fusion models, we used `dandelin/vilt-b32-mlm`[5] for ViLT and `ZinengTang/tvlt-base`[6] for TVLT. We have searched through various hyperparameter settings including batch size [16, 32, 64], learning rate [0.0001, 0.00003, 0.00001]. For multi-task learning, we considered weighing unimodal and multimodal objectives by factors of [0.1, 0.5, 1]. All models are trained on a NVIDIA A40 GPU.

---

[1]https://huggingface.co/

[2]https://huggingface.co/bert-base-uncased

[3]https://huggingface.co/google/vit-base-patch16-224

[4]https://huggingface.co/MIT/ast-finetuned-audioset-10-10-0.4593

[5]https://huggingface.co/dandelin/vilt-b32-mlm

[6]https://huggingface.co/ZinengTang/tvlt-base

