# OpenReview forum: "Is the Glass Half-Empty or Half-Full? A Mixture-Of-Tasks Perspective on Missing Modality"
_ICLR.cc/2024/Conference — ICLR 2024 Conference Withdrawn Submission_

### Official Review · Reviewer_PHyn · 2023-10-30

**Soundness:** 1 poor
**Presentation:** 2 fair
**Contribution:** 1 poor
**Rating:** 3
**Confidence:** 3

**Summary:**

The paper describes an evaluation scenario for multimodal models. The approach is described as a scenario where added modalities are seen as extra signals to improve performance.

**Strengths:**

The paper attempts to bridge a gap between practical use and academic evaluation of multimodal models. This is good. However, the distinction drawn between half-full and half-empty is not naturally obvious.

**Weaknesses:**

- Poorly organized and written. Difficult to clearly understand what the approach adds to current multimodal evaluation schemes.
- Experimental section is difficult to understand.

**Questions:**

**Suggestions**
- Please clarify what "modality competition" actually means or add reference early in the paper -- the term is used in both the abstract and introduction section.
- I suggest reorganization of the sections and rewriting most of the paper. It should be clearly stated, in the abstract, what the main issue is, what is presented as a finding/solution, and what is presented to support claims. In this paper, the abstract is too vague and imprecise.

**Questions**
- What is the main contribution of the paper? dropping 100% of one or two modalities has been used as an evaluation scheme in earlier works.
- Definition 2 is very specific (it is designed with two modalities). Do you have an idea of how this approach scales to $N$ modalities? Could the approach be facing a combinatorially large number of experiments?
- What is "MMBT" in the caption of Table 2?

---

### Official Review · Reviewer_FaQ6 · 2023-10-31

**Soundness:** 2 fair
**Presentation:** 1 poor
**Contribution:** 1 poor
**Rating:** 3
**Confidence:** 4

**Summary:**

Instead of formulating missing modality as a robustness task, authors argue  it as a multimodal representation learning task. A framework is proposed to construct systems that are robust to missing modalities. A benchmark for missing modality evaluation is proposed, and a wide variety of state- of-the-art models are compared.

**Strengths:**

This paper does not propose any new algorithm rather it just argues to consider missing-modality problem as a multimodal representation learning task. Recent works  such as `Learning Transferable Visual Models From Natural Language Supervision' have already demonstrated the advantages of natural language supervision in addition to the visual modality. Therefore a discussion on missing modalities appear to be outdated in the current context.

**Weaknesses:**

1. The writeup of this paper is quite different from normal research papers where something new is proposed and then advantage of the new invention is demonstrated using some applications.

2. The innovation of this work remains unclear. What new is proposed remains elusive.

3. The advantage of the proposed framework remains unclear.

4. A reader may not be able to dig what authors have proposed and proved in this paper.

5. A comparison of existing methods is given using existing datasets with an aim to infer something new. However, it remains unclear what new was inferred?

6. The authors does not consider the multimodal papers using natural language supervision. Missing modality problem has already been effectively addressed by multimodal works like CLIP. Semantic alignment of  textual and visual concepts in the embedding space has already addressed this problem.

**Questions:**

The same as weakness section.

---

### Official Review · Reviewer_qtFd · 2023-11-01

**Soundness:** 2 fair
**Presentation:** 2 fair
**Contribution:** 3 good
**Rating:** 3
**Confidence:** 2

**Summary:**

In this paper, the task of multi-modal learning is addressed, specifically methods of modality competition as well as robustness to missing modality. The authors assert that the way prior art measure these effects does not agree with actual use cases of multi-modal models and brings practical challenges. As declared by the authors, this is due to changing dynamics of experimental designs as well as infinite number of evaluation tasks due to sampling probabilities of modality loss.
To address these concerns, the authors propose their main contribution, an alternative approached called Missing Modality Performance Testbed (MMPT). In essence, it is a triplet of values that measure the performance of a given model with respect to a metric on the full test set and the test sets without their respective modalities. The next contribution is demonstrating how to decouple early fusion transformer model into their constituent unimodal parts for the aforementioned proposed evaluation.
In the experimental results section, the authors aggregate results across the related literatures of missing modality and modality competition, both for early- and late-stage fusion. In addition, they demonstrate their proposed MMPT and showed how the model separation - which is a requisite for MMPT - can improve overall performance and robustness to missing modalities. This constitutes their final contribution.

**Strengths:**

- Accumulate results across related literature. This is valuable for the community as an overview how methods from related fields perform on multiple different datasets This is important, as it captures the current state of multi-modality training and its current problems with respect to missing modality / modality competition
- Demonstrated a method to disentangle a early-fusion transformer into its constituent unimodal components
- Demonstrated improved performance on their benchmark using existing work by separating the models into their constituent parts and evaluating the system based on the best performing constituent (e.g either unimodal model or the multi-modal model).

**Weaknesses:**

- Many claims the authors state is not backed up or adequately motivated with references or arguments. This includes the premise. This weakens the contributions of the paper as it is not evident why MMPT is needed:
    - Why is unimodality for machines to be expected? (Sec. 1)
    - Why is there no universally applicable way to compare proposed methods? (Sec. 1)
    - I understand that evaluating on all p in the interval [0,1] is infeasible, but why does it not suffice to subsample that space to give evidence on how models perform under varying probabilities for modality drop? (Sec. 1, 2.1)
   - Why is achieving a higher performance on one modality potentially bad for the other? (Sec. 1)
    - Is there any evidence that a model could sabotage itself to increase the robustness score? (Sec. 2.1)
I'm not arguing against the validity of these statements, but merely expressing my concern over the lack of backing evidence.
- Sec 2.2 introduces MMPT, and claims that it captures any combination of training and testing probabilities. Yet there is no empirical evidence provided that this is indeed the case. Therefore it is not clear to me that this claim holds.
- Sec 2.2 demonstrates a use-case of MMPT, but does not introduce the model or the task at hand. This makes understanding the figure more challenging and time consuming.
- Tbl. 1 and 2 are difficult to parse. Why is the best performing model highlighted for each modality? Do we not care about the overall best performing method given a certain test set? It is also not clear what the blue background means, as the textual explanation does not align with what is highlighted. (e.g on MM-IMDB dataset, OGM-GE outperforms its unimodal expert but is not highlighted in blue)
- Sec. 4.2, mentions interesting further considerations when selecting a model apart from just its accuracy. Yet it is only informally stated in the text with no tables or figures provided. For example, it would have been nice to have a table of training speeds, number of trainable parameters, or seeing metrics on training stability. This would have increased the value of this paper to the community, as these are values that are usually under-reported in literature.

In summary, the paper is difficult to read due to the lack of backing and explanation of statements the authors make in the paper. There is missing details (e.g task and model of Fig. 2). The main contribution, MMPT, has not been empirically validated. Due to these points, it makes it difficult for me to assess the usefulness of the main contribution.

**Questions:**

- Why is the weakness score defined in terms of loss (smaller is better), whereas MMPT is defined in terms of an evaluation metric where higher is better? Wouldn't picking one or the other make both definition more consistent with each other?
- Does the modification of the self-attention mechanism introduce a performance penalty on the transformer models?
- Sec. 3.2 (towards the end), shouldn't psi_{m_i} be set to z_{m_i}?
- Sec. 4.1, naive joint training baseline, how are the representations fused exactly?

---

### Official Review · Reviewer_6Lgm · 2023-11-01

**Soundness:** 2 fair
**Presentation:** 3 good
**Contribution:** 1 poor
**Rating:** 3
**Confidence:** 3

**Summary:**

This paper investigates the effect of missing modality in multimodal learning and proposes an approach called the "Missing Modality Performance Testbed" (MMPT). The proposed method also connects missing modality and model competition that aims to improve the unimodal representations for late-fusion in multimodal settings.

The main contribution of the paper is an evaluation strategy that compares different fusion strategies. Each system is trained don aggregated unimodal and multimodal settings and split into unimodal $(\psi^{m_1}, \psi^{m_2})$ and multimodal component $(\psi^{m_1, m_2})$. Later, these components are linear-probed on respective modalities in the test set. Multimodal strategies are compared to the unimodal by relating each other using "robustness to missing modality" and "weakness score".

All experiments are conducted on 6 two-modality databases including text, image, and audio (MM-IMDB, N24News Headline, N24News Body, Crisis-MMD, UPMC Food101 and AV-MNIST).

**Strengths:**

1. Several early- and late-fusion methods are compared on six multimodal databases. The extensive experiments give insights into the performance of each modality and the fusion method.

2. These experiments help the reader to see joint multimodal (Naive) and Unimodal training with several approaches, including Dropout (randomly imputing a modality with a probability of $p$ at training time). As stated in the paper, preferred model architectures are all state-of-the-art transformer models (BERT, ViT, and AST); thus, early- and late-fusion experiments give a realistic result of these strategies.

**Weaknesses:**

1. *Regarding the comparison between human and machine decision-making on Page 2:* In both human and machine cases, multimodality is expected as many concepts that we aim to learn under supervised learning settings depend on multiple modalities. Unimodality in human decision-making is not always unfavorable.

2. *The analogy to glass-half and glass-empty not clear:* Tables 1 and 2 using Macro F1 scores in test sets. In different training schemes, the performance of two modalities and fusion scores were reported. I think "glass-half-full" as in Definition 2, is to write all performance metrics (F1 scores) side-by-side, and "glass-half-empty" is the comparison of an unimodal setting over a multimodal case. Here, the structural patterns on results tell about the most contributing single modality and fusion strategy. In any ML papers using multiple modalities, we see a similar comparison.

3. Figure 3 reports the best fusion method and compares it with Ma et al. (2022), Lee et al. (2022), and Lee et al. (2023). However, it is unclear why the authors made this comparison because 5-6 methods (including Ma et al. (2022)) are used in the paper. When one picks the model with the best F1 score among several fusion strategies, it will be always better than these approaches.
    * A metric calculated during training phase that can automatically weights or picks the best fusion sample-by-sample could be regarded a good comparison. However, the proposed approach is only combination of performance metrics, i.e., F1 scores (glass-half-full), and the difference of single modality vs. multimodal performance metrics (glass-half empty).

**Questions:**

Please see my comments under Weaknesses section.

My major concerns and questions are as follows:
* Looking into the previous work (for instance, modality competition in Huang et al. 2022) already reported theoretical and experimental findings that multimodal, jointly learned representations are unsatisfactory and impairs the expertise in the unimodal case. So, the unimodal performance's being powerful in most experiments in Tables 1 and 2 is not surprising. What is the contribution of MMPT (which is not a different performance metric than the primarily used one)?

* The proposed metric is not a novel approach, it simply compares unimodal and multimodal approaches either by the difference or ratio of their performance metrics. Aside from the novelty of the MMPT, it requires several fusion strategies applied to compare.

Overall, comparing several early- and late fusion strategies and recent approaches is valuable, however, the evaluation is straightforward and does not introduce a better perspective from benchmarking perspective and I would like the authors elaborate the usefulness of proposed approach different than the recently published missing modality and modality competition papers.

**Details Of Ethics Concerns:**

No ethics review needed. There is no human data used, and the topic of the paper does not contain direct privacy, security or fairness concerns.